# Progression-Dependent Altered Metabolism in Osteosarcoma Resulting in Different Nutrient Source Dependencies

**DOI:** 10.3390/cancers12061371

**Published:** 2020-05-27

**Authors:** Raphaela Fritsche-Guenther, Yoann Gloaguen, Marieluise Kirchner, Philipp Mertins, Per-Ulf Tunn, Jennifer A. Kirwan

**Affiliations:** 1Berlin Institute of Health Metabolomics Platform, Berlin Institute of Health (BIH), 13125 Berlin, Germany; yoann.gloaguen@mdc-berlin.de (Y.G.); Jennifer.Kirwan@mdc-berlin.de (J.A.K.); 2Max Delbrück Center for Molecular Medicine (MDC) in the Helmholtz Association, 13125 Berlin, Germany; marieluise.kirchner@mdc-berlin.de (M.K.); Philipp.Mertins@mdc-berlin.de (P.M.); 3Core Unit Bioinformatics, Berlin Institute of Health (BIH), 10178 Berlin, Germany; 4Proteomics Platform Berlin Institute of Health (BIH) and Max Delbrück Center for Molecular Medicine (MDC) in the Helmholtz Association, 13125 Berlin, Germany; 5Department of Orthopedic Oncology, Helios Clinic Berlin-Buch, 13125 Berlin, Germany; per-ulf.tunn@helios-gesundheit.de

**Keywords:** osteosarcoma, GC-MS, flux analysis, glucose, glutamine, sex and gender

## Abstract

Osteosarcoma (OS) is a primary malignant bone tumor and OS metastases are mostly found in the lung. The limited understanding of the biology of metastatic processes in OS limits the ability for effective treatment. Alterations to the metabolome and its transformation during metastasis aids the understanding of the mechanism and provides information on treatment and prognosis. The current study intended to identify metabolic alterations during OS progression by using a targeted gas chromatography mass spectrometry approach. Using a female OS cell line model, malignant and metastatic cells increased their energy metabolism compared to benign OS cells. The metastatic cell line showed a faster metabolic flux compared to the malignant cell line, leading to reduced metabolite pools. However, inhibiting both glycolysis and glutaminolysis resulted in a reduced proliferation. In contrast, malignant but non-metastatic OS cells showed a resistance to glycolytic inhibition but a strong dependency on glutamine as an energy source. Our in vivo metabolic approach hinted at a potential sex-dependent metabolic alteration in OS patients with lung metastases (LM), although this will require validation with larger sample sizes. In line with the in vitro results, we found that female LM patients showed a decreased central carbon metabolism compared to metastases from male patients.

## 1. Introduction

Osteosarcoma (OS) is a primary malignant bone tumor that usually develops in children and young adults during periods of rapid growth [1]. Tumors are mainly located in the metaphyseal regions of long bones. OS most frequently occurs in the second decade of life, with about 60% of patients under 25 and 30% over 40 years. Evidence suggests that OS arises from transformed mesenchymal cells [2]. The tumor is termed “primary” when the underlining bone is normal and “secondary” when an underlying cause has been identified—e.g., the bone has been damaged due to conditions like irradiation, Paget disease, infarction or other disorders. OS is the most common radiation-induced sarcoma [2].

OS metastases are mostly found in the lung (85–90%) and are still the primary cause of death in OS patients [3]. Nearly all patients develop chemotherapy-resistant lung metastases; the 5-year overall survival rate remains at about 70% [4]. Current treatment options for OS involve surgery combined with chemotherapy. Unfortunately, the treatment of OS patients has remained unchanged since the 1970s [4,5]. The limited understanding of the biology of metastatic processes in OS limits the ability for effective treatment.

Metabolomics is a relatively new discipline to gain insight into the ongoing metabolism of biological specimens. Metabolites are an objective measurement of the phenotype of a cell, and measure the interaction of the genotype and environment of a cell [6]. Thus, metabolites may serve as biomarkers for cancer diagnosis and/or prognosis. To date, metabolomic signatures have already been published for several epithelial cancers, such as colon, liver, lung, pancreatic, ovarian and breast [6,7]. Previous research in OS is more modest and has focused on understanding the pathogenesis and development of OS. Metabolic alterations have previously been used to classify OS tumors [8,9,10,11,12,13,14,15,16]. However, there are still unanswered questions relating to the metabolic transformation during metastasis. A better understanding of the global metabolic alterations is of great importance to enhance OS therapy.

The current study intended to identify metabolic alterations during OS progression by using a combined metabolomics and proteomics approach. We aimed to elucidate how such metabolic changes are advantageous to the survival of malignant OS cells. We sought to determine the metabolic pattern using a cell-based progression model of human OS as well as primary and lung metastases tissue from OS patients. In addition, we specifically investigated whether there were sex-dependent differences in cell metabolism.

## 2. Results

### 2.1. Enhanced Metabolism in Malignant and Metastatic OS Cells

A clinically relevant model is important for elucidating the potential role of metabolites involved in the progression of OS. There exists a progression-based model of human OS which encompasses three related cell lines: (1) the benign human osteosarcoma (HOS) HOS-TE85 clone F5 cell line, (2) the chemically transformed (N-methyl-N’-nitro-N-nitrosoguanidine = MNNG) malignant MNNG/HOS cell line and (3) the kirsten rous sarcoma (KRAS) transformed 143B cell line, which are all derivatives from the same patient, suggesting a similar genetic background [17]. All three cell lines showed varying tumor growth capabilities and spontaneous metastatic potential [17]. The parental HOS cell line was cultured from a 13-year-old Caucasian female patient. The 143B cell line exhibits the greatest cell motility and invasion capability and was able to grow anchorage-independent [17,18]. When these three cell lines were orthotopically injected into nude mice, both the MNNG/HOS and 143B cells were able to form tumors with high efficiency, while HOS did not. Only the 143B-injected animals developed numerous pulmonic metastases. Therefore, the three cell lines were used to investigate the different stages of OS development and progression.

A gas chromatography-mass spectrometry (GC-MS) approach was employed to identify pre-defined metabolites associated with the central carbon metabolism (CCM). This includes the basic energy pathways of glycolysis, the tricarboxylic acid (TCA) cycle and amino acids (AA). The comparison of the benign HOS to the malignant MNNG/HOS and metastatic 143B cell line revealed an upregulated CCM (Figure 1A, Appendix A). The comparison of the malignant MNNG/HOS to the metastatic 143B cell line showed decreased pools of the metabolites from the CCM in the 143B cells. To analyze the general effect on energy metabolism, the data were treated as follows. First, the results for each metabolite across all cell types were univariate scaled [19]. This had the advantage of giving all the metabolites a common non-parametric scale for comparison. Next, the metabolites were grouped according to their metabolic class (e.g., glycolysis, TCA cycle or amino acids), and the results for all the metabolites per class were considered together for analysis by plotting them on a common axis and assessing trends for increases or decreases per biological class. The combined analysis revealed a significant increase in metabolites from glycolysis, the TCA cycle and amino acids in MNNG/HOS cells compared to HOS cells (Figure 1B). The comparison of 143B to the HOS cells revealed a significant increase in the metabolites from the TCA cycle and the amino acids in the 143B cells. A significant decrease in metabolites from the TCA cycle and amino acids could be shown for the 143B cells in comparison to the MNNG/HOS cells.

Next, proteomic profiling was carried out on the three OS cell lines. The analysis revealed 3626 significantly differentially regulated proteins when comparing the HOS and MNNG/HOS cells, 3641 comparing the 143B and HOS cells and 4096 between the 143B and MNNG/HOS cells. The overlap is shown in Appendix A. In line with the results from our metabolic approach, most of the proteins associated with glycolysis were significantly upregulated in MNNG/HOS and 143B when compared to the HOS cells, while they were downregulated in 143B compared to the MNNG/HOS cells (Figure 1C). For the TCA-related proteins, a similar regulation was found in the MNNG/HOS and 143B cells compared to the HOS cells, while a reciprocal regulation was found in the 143B cells.

By analyzing the lipid and polar metabolites using a kit-based approach, similar changes were observed as for the CCM and proteins (Figure 2 and Appendix A). Most of the compounds had increased in the MNNG/HOS and 143B cells compared to the HOS cells. In the 143B cells, a decrease in nearly 50% of the compounds was found compared to the MNNG/HOS cells.

To analyze whether the decreased CCM metabolites in the 143B cells came from lower production or increased usage, we performed a pulse stable isotope time-resolved metabolomics (pSIRM) approach. In the presence of ^13^C-glucose or ^13^C-glutamine, we traced the incorporation of the labelled carbons into glycolysis or TCA cycle intermediates, respectively. The comparison of the HOS and MNNG/HOS cells revealed no major differences within the ^13^C-glucose label incorporation into the glycolytic metabolites (Figure 3A and Appendix A). By contrast, a higher glycolytic flux could be shown in the 143B cells compared to the HOS or MNNG/HOS cells. Interestingly, serine showed a higher label incorporation in the 143B cells compared to the HOS and MNNG/HOS cells. Therefore, we analyzed the expression of the serine pathway proteins D-3-phosphoglycerate dehydrogenase (PHGDH) and phosphoserine aminotransferase (PSAT1) (Figure 3B). Since a recent study showed a direct link between serine/glycine metabolism and the mechanistic target of rapamycin complex 1 (mTORC1), leading to proliferation and survival in the MG63 OS cell line, we analyzed the level of activated mechanistic target of rapamycin (mTOR) in our OS cell progression model 24 hours (h) after cell plating. As shown in Figure 3C, mTOR was highly activated in the MNNG/HOS cells compared to the HOS and 143B. To understand better the proliferation potential of the three cell lines, the phosphorylation of the extracellular signal-related kinases 1 and 2 (ERK1/2) was analyzed. ERK1/2 have an important role in cell mitosis, and the phosphorylated form of the kinase is the active form. Phosphorylated ERK1/2 showed elevated levels in the MNNG/HOS and 143B cells compared to the HOS, suggesting a higher proliferative capacity (Figure 3D).

After feeding the cells with ^13^C-glutamine, an increased label incorporation into the TCA metabolites was found in the 143B cells compared to the HOS and MNNG/HOS cells (Figure 3E and Appendix A). Additionally, a higher label incorporation into 2-hydroxy-glutaric acid (2HG) was found for 143B compared to the MNNG/HOS and HOS cells (Figure 3F). Due to high level of label incorporation into citric acid, we suggest that TCA metabolites are being produced from glutamine via the reverse citric acid cycle.

### 2.2. Progression-Dependent Switch in Nutrient Sources

Since we found an increased glycolytic flux in the MNNG/HOS and 143B cells compared to their benign counterpart, we were interested in the ability of the cells to deal with perturbations in the CCM. Cell proliferation and increase in cell mass is highly dependent on glucose metabolism [20,21]. It has been suggested that the metabolites of glycolysis play an important role in cell growth by activating the mTOR complex [22]. However, enhancing glucose utilization alone does not drive cell growth, suggesting that it is unlikely that glucose or glycolytic metabolites directly stimulate mTOR and therefore proliferation. The reliance on glutamine has long been considered a hallmark of cancer cell metabolism; however, nowadays there is evidence that several factors, like tissue type or micro environmental parameters, can influence glutamine dependency. Therefore, we analyzed the capability of the OS cells to deal with the starvation of glutamine or glucose. We found that all three cell lines showed a slight but non-significant reduction in the viability of the cells when the glucose was removed. The deprivation of glutamine induced a cell death in the HOS and MNNG/HOS cells, while the 143B cells were unaffected (Figure 4A).

It is already known that the glycolysis and glutaminolysis of cancer cells can be blocked by using glycolytic inhibitors like 3-Bromopyruvic acid (BrPy) [7,23]. Therefore, we assessed the viability of the OS cell lines treated with BrPy. As predicted from the analysis of the phosphorylation state of ERK, the MNNG/HOS and 143B cells showed a higher proliferation after 24 hours (h) compared to the HOS cells (1.84 and 2.22-fold increase in the number of viable cells, respectively) (Figure 4B and Appendix A). After 72 h, the proliferation of the HOS and 143B cells was equal (3.86 and 3.35-fold increase in the number of viable cells compared to 0 h, respectively), while the MNNG/HOS cells showed the highest proliferative rate (6.72-fold increase in the number of viable cells compared to 0 h). The inhibitor reduced the growth of the HOS and 143B cells (1.75-fold decrease and 1.69-fold decrease after 72 h compared to 0 h, respectively), but by contrast, the MNNG/HOS cells were able to proliferate (2.45-fold increase after 72 h compared to 0 h), although an initial inhibition in growth was seen even in these cells for the first 48 h following treatment. An increased phosphorylation of mTOR was shown in the HOS and MNNG/HOS cells after BrPy treatment for 24 h, while phosphorylation decreased in the 143B cells (Appendix A).

### 2.3. Sex-Dependent Metabolic Alteration in OS Patients with Primary Tumors and Lung Metastases

Seven primary tumors (PT) and five lung metastases (LM) from different individuals were analyzed. Hierarchical clustering of the samples revealed a split of the lung metastases into two groups (Figure 5A). An examination of potential confounding factors showed a separation according to the different sexes; we were cautious in our interpretation of these data, since only *n* = 2 male and *n* = 3 female LM were available. The female LM clustered together, while the male LM were more heterogeneous in their clustering. Since only one female patient was available in the PT group and her results clustered together with the PT males, the following data analysis was performed by averaging them. The LM were separately analyzed according to sex.

If we consider the average normalized area of the metabolite groups based on key metabolic pathways, the male LM samples showed a non-significant trend towards a slightly higher CCM intensity compared to the PT reflected by glycolysis, the TCA cycle, glycerol pathway, pentose phosphate pathway and amino acids (Figure 5B and Appendix A). When analyzing the female LM samples, a significant decrease in the TCA cycle (4/4), glycolysis (2/3) and amino acid pools (13/16) were observed compared to the PT.

## 3. Discussion

The process of the metastatic spread of tumors to distant organs is poorly understood. The cells need to overcome numerous hurdles. To successfully metastasize, a tumor cell must survive the barriers that inhibit movement out of the primary tumor, enter into the circulation and re-establish themselves within a distant organ. This makes metastasis a highly inefficient process. The study of cancer cell metabolism has improved our understanding of carcinogenesis and cancer progression and provided knowledge to develop therapeutic approaches. A systematic characterization of the metabolic pathways in OS is lacking, and the contribution of these metabolic alterations in promoting OS metastatic development is unknown. In this work, we focused on metabolic differences between benign, malignant and metastatic OS cells to gain more insights into the biology of metastasis. In addition, we applied a metabolic screening of patient material to describe metabolic changes from primary tumors and lung metastases using a targeted GC-MS, liquid chromatography mass spectrometry (LC-MS) and kit-based (polar and lipid compounds) approach.

Compared to non-transformed cells, cancer cell metabolism is reprogrammed to support accelerating proliferation and to adapt to the tumor environment [24]. The same needs exist for metastatic cells. Several studies showed that metastasizing cells become dependent on glycolysis for adenosine triphosphate (ATP) production [25]. The in vitro OS cell line model demonstrated an enhanced metabolism in malignant and metastatic OS cells compared to benign cells. The metastatic cell line showed a faster metabolic flux in the glycolysis and TCA cycle compared to the malignant cell line when glucose or glutamine was used as a nutrient source, leading to reduced pools of CCM. Ribose-5-phosphate showed a reduced label incorporation in both malignant and metastatic cells compared to benign cells, suggesting a decreased flux through the pentose phosphate pathway. This is similar to the metabolic changes we saw in our clinical samples from female patients.

Normal cells differ from cancer cells by changes in their energetic and metabolic properties. Cancer cells are more dependent on glycolysis, glutaminolysis and fatty acid synthesis to sustain the higher proliferation rate [26,27]. In cancer cells, glycolysis is increased even in the presence of oxygen [8]. However, glutaminolysis is also strongly increased in cancer cells as an alternative pathway for energy production [28]. Glutaminolysis provides the carbon source for the TCA cycle, which is highly essential for several biosynthetic pathways. Glutamine is a precursor of alpha-ketoglutaric acid (αKG), which enters the TCA cycle and increases the production of oxaloacetate reacting with acetyl-Coenzyme A to produce citric acid. However, especially in tumor cells, glutamine also contributes to the reductive carboxylation of isocitrate dehydrogenase (IDH). It is already known that 143B cells show a suppression of mitochondrial function [29]. Due to the importance of glutaminolysis, most cancer cells are sensitive to glutamine deprivation and cannot proliferate in culture medium without glutamine. We found that, independent from the tumor progression, the OS cells undergo a switch from using glucose to glutamine when no glucose is provided. When glutamine is limited, a decreased proliferation in the HOS and MNNG/HOS cells was found. In contrast, the 143B cells switched to using glucose when glutamine was limited.

It is already known that targeting amino acid metabolism in cancer is a promising strategy for the development of novel therapeutic agents. Amino acids are essential to support the high metabolic demands of tumor cells to deal with the conditions of the tumor microenvironment. In particular. metastatic cells need to deal with certain stressful conditions in the tumor microenvironment. The amino acid arginine is an important precursor for the synthesis of proteins, urea and creatinine and for the synthesis of glutamate, nitric oxide and agmatine [30]. A lot of tumors depend on exogeneous arginine for their growth, since they lack argininosucchinate synthetase 1 (ASS1) [31]. OS patients lacking ASS1 correlate with the development of pulmonary metastases [32]. Serine and glycine metabolism are interconnected via the glycine cleavage system, a major metabolic pathway in one-carbon metabolism that provides cofactors for pyrimidine and purine nucleotide biosynthesis. Cancer cells use both amino acids to synthesize building material for cell growth and proliferation [33]. The serine/glycine metabolism controlled by the mTORC1 pathway act as a protective system in OS cells [34]. This metabolic signaling promotes OS cell proliferation and the ability to deal with micro-environmental stress, leading to enhanced OS cell survival. It is known that cancer cells have a higher capacity for de novo serine synthesis via the PHGDH pathway. The serine biosynthetic pathway is upregulated in highly metastatic breast cancer cells and associated with poor survival [35]. In our study, glycine and serine showed significantly lower pools in female OS LM compared to male LM and PT and in the counterpart in vitro cellular models 143B compared to MNNG/HOS. However, the in vitro flux analysis revealed a high label incorporation into serine. In accordance, a significant increase in the serine pathway proteins were found, suggesting that the serine and glycine pools were decreased due to their increased use as energy sources.

Metastatic OS cells rely on glutamine as a key source for proliferation [36]. Pharmacological inhibitors of glycolysis or oxidative phosphorylation have previously been used to explore whether perturbing cancer energy metabolism is a potential treatment target. BrPy is an agent blocking ATP synthesis by the inhibition of glycolysis and the mitochondrial electron transport chain (ETC), resulting in the limited growth of many tumors with no effect on non-transformed cells. It was also shown that BrPy is not able to directly inhibit the activity of glutaminolytic enzymes; however, in the TCA cycle, BrPy is able to inhibit IDH and αKG activities, leading to a decreased glutamine metabolism due to the importance of IDH and αKG activities in incorporating αKG derived from glutaminolysis into the TCA cycle [37]. Blocking glycolysis and glutaminolysis led to cell death in 143B metastatic OS cells due to the inhibition of all available nutrients. Similar effects could be shown in human MG63.3 and murine K7M2 highly metastatic OS cells after glucose and glutamine deprivation [36]. Interestingly, the MNNG/HOS cells were able to recover after 72 h of BrPy treatment, suggesting a currently unknown metabolic circumvention mechanism.

Our in vivo metabolic approach hinted that there may be sex-dependent metabolic alteration in OS patients with LM. In our study, the female LM patients showed decreased central carbon metabolites compared to metastases from male patients, but we recognize that with so few samples, further follow up with a larger sample size is required. Zhang et al. have shown that amino acids and the gluthathione and polyamine metabolism are upregulated in OS patients compared to healthy donors [38]. A metabolomics study using a mouse model with the subcutaneous transplantation of the murine OS cell line LM8 revealed an upregulated metabolism during lung metastasis [9]. In the study of Ren et al., functional changes in the cellular metabolism of highly metastatic cells were found when compared to cells with a low metastatic potential [15]. The metabolites involved in arginine metabolism, glutathione metabolism, lipid metabolism and the inositol pathway were identified in highly metastatic OS cells compared to their low metastatic parental cells. However, none of the studies considered sex as a parameter for changes in the metastatic progression. Since we found sex-specific differences in our in vivo analysis, we had a closer look to the sex of the established OS progression model. We determined that the parental HOS cells were generated from a 13-year-old female patient. The female in vitro OS cell line model demonstrated that the metastatic cell line showed a faster metabolic flux compared to the malignant cell line. This was similar to the metabolic changes we saw in our clinical samples from female patients.

OS is one of the most complex oncological diseases in terms of genetic aberrations [39,40]. Tumor heterogeneity relies on the cancer cells and the tumor microenvironment, which is composed of different cells types including immune cells, mesenchymal stem cells, endothelial cells and fibroblasts. To reduce the genetic variation for data analysis, a progression model was used that was derived from a single cell line. To statistically verify the obtained results and evaluate the sex-specific differences in progression states would ideally require more of these progressive OS cell lines from different sexes. Further follow up would also allow us to better understand the effects that tumor heterogeneity may exert on the results.

In summary, in the present study progression-dependent metabolic differences were found in vivo and in vitro, leading to the assumption that female metastatic cells have a faster metabolism and mainly use one-carbon and glutamine as an energy source compared to malignant cells. The unavailability of glucose or glutamine can be circumvented via switching from glycolysis to glutaminolysis and vice versa. However, the cells were as sensitive to the inhibition of glycolysis and glutaminolysis as their non-malignant counterparts. By contrast, the malignant OS cells were dependent on glutamine but performed a switch from glycolysis to glutaminolysis when starved of glucose. Interestingly, the malignant OS cells showed a time-dependent proliferation recovery after treatment with small molecule inhibitors of glycolysis and glutaminolysis. The mechanism of this recovery and which metabolic pathways are involved in it are still unknown. Finally, our results may provide new opportunities for drug development in OS clinical therapy and show the importance of considering the sex and progression specificity of OS cells. However, further follow up with a larger patient sample size and multiple male and female progression-dependent OS cell line models is required.

## 4. Materials and Methods

### 4.1. Sample Collection

OS tissue including PT and LM from 11 patients was collected from freshly isolated resections at Charité/Robert-Rössle-Clinic Berlin Buch, Germany. Specific information about the tumor type, age, sex, grade, chemotherapy and acquired metastases is shown in Table 1. Informed consent was obtained from each patient before sample collection and the procedures were approved by the institution. All the subjects gave their informed consent for inclusion before they participated in the study. The study was conducted in accordance with the Declaration of Helsinki, and the protocol was approved by the Ethics Committee of the Robert-Rössle-Clinic Berlin Buch (AA3/03/45). The tissue specimens were dissected and snap-frozen in liquid nitrogen and stored at −80 °C. The tissue samples were used for GC-MS metabolomic profiling. The inclusion criterion was a histo-pathological diagnosed OS.

### 4.2. Cell Lines

The OS cell lines HOS and MNNG/HOS were obtained from ATCC (American Type Culture Collection, Manassas, VA, USA). The 143B cell line was purchased from Sigma-Aldrich (St. Louis, MO, USA). The cell lines were maintained in RPMI1640 (Merck Millipore Burlington, MA, USA) supplemented with 10% fetal calf serum (FCS, Thermo Fischer Scientific, Waltham, MA, USA) and 1% penicillin/streptomycin (Thermo Fischer Scientific, Waltham, MA, USA). All the cells were incubated in a humidified atmosphere of 5% CO_2_ in air at 37 °C. For glucose and serum starvation, RPMI1640 without glucose and glutamine was used and supplemented either with 2 g/L of glucose or 2 mM of glutamine.

### 4.3. Inhibitor Studies

For inhibitor studies, 24 h after plating inhibitor was added at the indicated times. The cells were counted with a TC20 cell counter (BioRad, Hercules, CA, USA) using Trypan blue for the detection of viable cells. The cell pellet was weighed, snap frozen in liquid nitrogen and stored until analysis at −20 °C. The glycolysis/glutaminolysis inhibitor 3-bromopyruvate (Sigma-Aldrich, St. Louis, MO, USA) dissolved in phosphate buffered saline (PBS) was added from a sterile stock solution to a final concentration of 0.25 mM.

### 4.4. Cell Proliferation Measurement Using XTT

Forty-eight hours after changing the media which was supplemented with glucose and glutamine or starved of either glucose or glutamine, the XTT (2,3-Bis-(2-methoxy-4-nitro-5-sulfo phenyl)-2H-tetrazolium-5-carboxanilid) reagent (Abcam, Cambridge, UK) was added in a ratio of 1:1. The XTT reagent consisted of a 600 µL electron coupling reagent and 600 µL of XTT. After 3 h of incubation at 37 °C and 5% CO_2_, the absorbance was measured at 450 nm using an Infinite M200Pro plate reader (TECAN, Männedorf, Switzerland).

### 4.5. Luminex Bead Based Analyis

The cell lysates were collected and the level of phospho-protein expression was analyzed with the MagPix system (Merck Millipore Burlington, MA, USA). The following multiplex kits were used according to the manufacturer’s instructions: Milliplex MAP Akt/mTOR phosphoprotein magnetic bead 11-plex kit, Milliplex MAPK/SAPK signaling 10-Plex kit and Milliplex map β-Tubulin total magnetic beads (all Merck Millipore Burlington, MA, USA). Briefly, the cell pellet was lysed with a cell lysis buffer (Merck Millipore Burlington, MA, USA). The lysate protein concentration was determined with the BCA (bicinchoninic acid) method (Thermo Fischer Scientific, Waltham, MA, USA). A total of 15 µg was used for a multiplex analysis. For acquiring data, the xPONENT 4.2 software (Merck Millipore Burlington, MA, USA) was used.

### 4.6. Metabolomics Analyses

#### 4.6.1. Tissue Extraction

For each sample, 50 mg of OS tissue was pulverized using the CP02 pulverizer (Covaris, Brighton, UK). The tissue was lysed by the addition of 1 mL/50 mg tissue methanol (MeOH): chloroform (CHCl_3_):water (H_2_O) (5:2:1, *v/v/v*) mixture, followed by incubation for 30 min (min) at 4 °C with shaking. Next, 500 µL/50 mg of tissue of H_2_O was added, followed by incubation for 15 min at 4 °C with shaking. The vials were centrifuged at 18,213× *g* at 4 °C. After extraction, 500 µL of polar phases was dried at 30 °C at a speed of 1550× *g* at 0.1 mbar using a rotational vacuum concentrator (RVC 2-33 CDplus, Christ, Osterode am Harz, Germany). The samples were pooled after extraction to make a representative quality control (QC) sample to test the technical variability of the instrument. QC samples were prepared alongside the samples in the same way.

#### 4.6.2. Cell Culture Harvest and Extraction

The cells were seeded in an appropriate density for 48 h. Three replicates per time point and cell line were plated. For labeling studies, 4 h before harvest the media was changed to maintain the glycolytic activity. Afterwards, the cells were exposed for the indicated time to media containing 2 g/L of ^13^C-glucose (U13C6 D-glucose, Cambridge Isotope Laboratories, Tewksbury, MA, USA) or 2 mM of ^13^C-glutamine (13C5 L-glutamine, Cambridge Isotope Laboratories, Tewksbury, MA, USA). The cells were rapidly washed (20 seconds) with a wash buffer (140 mM NaCl, 5 mM HEPES), at pH 7.4, 37 °C, for the labeling experiments. Either a buffer containing ^13^C-glucose/^12^C-glutamine, ^12^C-glucose/^13^C-glutamine or ^12^C-glucose/^12^C-glutamine (zero time point) at the same concentration as in the original media was added for labeling before they were quenched with 5 mL of ice-cold 50% MeOH solution containing 2 µg/mL of cinnamic acid (for use as an internal standard). Immediately upon the MeOH solution being added to the culture plate, the cells were scraped into the methanol solution and the methanolic lysates were collected. After the cell harvest, 4 mL of CHCl_3_, 1.5 mL of MeOH and 1.5 mL of H_2_O were added to the methanolic cell extracts, which were shaken for 60 min at 4 °C and centrifuged at 4149× *g* for 10 min to separate the phases. The polar phase (6 mL) was collected and dried at 30 °C at a speed of 1550× *g* at 0.1 mbar using a rotational vacuum concentrator (RVC 2-33 CDplus, Christ, Osterode am Harz, Germany). The samples were pooled after extraction and used as a quality control sample to test the technical variability of the instrument. They were prepared alongside the samples in the same way. To generate backup samples, 20% MeOH was added to the dried extracts, which were shaken for 60 min at 4 °C and centrifuged at maximum speed (18,213× *g*) for 10 min. Two times, 280 µL was aliquoted and dried under vacuum.

After the collection of the polar phases, proteins were extracted for each sample by the addition of 8 mL of 100% MeOH, followed by centrifugation at maximum speed for 10 min. The supernatant was carefully discarded. The pellet was air dried and used for total protein lysis and protein determination.

#### 4.6.3. GC-MS Metabolomics Measurement of Key Central Carbon Pathway Metabolites

All the polar cell extracts were stored dry at −80 °C until analysis. The extracts were removed from the freezer and dried in a rotational vacuum concentrator for 60 min before further processing to ensure there was no residual water which may influence the derivatization efficiency. The dried extracts were dissolved in 15 µL of methoxyamine hydrochloride solution (40 mg/mL in pyridine) and incubated for 90 min at 30 °C with constant shaking, followed by the addition of 50 µL of N-methyl-N-[trimethylsilyl]trifluoroacetamide (MSTFA) and incubated at 37 °C for 60 min. The extracts were centrifuged for 10 min at 18,213× *g*, and aliquots of 25 µL were transferred into glass vials for GC-MS measurements. An identification mixture for reliable compound identification was prepared and derivatized in the same way, and an alkane mixture for a reliable retention index calculation was included [7,23,41]. The metabolite analysis was performed on a Pegasus 4D GCxGC TOFMS-System (LECO Corporation, St. Joseph, MN, USA) complemented with an auto-sampler (Gerstel MPS DualHead with CAS4 injector, Mühlheim an der Ruhr, Germany). The samples were injected in split mode (split 1:5, injection volume 1 µL) in a temperature-controlled injector with a baffled glass liner (Gerstel, Mühlheim an der Ruhr, Germany). The following temperature program was applied during the sample injection: for 2 min, the column was allowed to equilibrate at 68 °C, then the temperature was increased by 5 °C/min until 120 °C, then by 7 °C/min up to 200 °C, then by 12 °C/min up to a maximum temperature of 320 °C, which was then held for 7.5 min. The gas chromatographic separation was performed on an Agilent 7890 (Agilent Technologies, Santa Clara, CA, USA), equipped with a VF-5 ms column (Agilent Technologies) of 30 m length, 250 µm inner diameter and 0.25 µm film thickness. Helium was used as the carrier gas with a flow rate of 1.2 mL/min. The spectra were recorded in a mass range of 60 to 600 m/z with 10 spectra/second. The GC-MS chromatograms were processed with ChromaTOF software (LECO Corporation, St. Joseph, MN, USA), including a baseline assessment, peak picking and computation of the area. Forty-five targeted metabolites from the CCM were selected for further data analysis (Appendix A).

#### 4.6.4. Protein Determination

The protein amount was determined using the BCA (bicinchoninic acid) method (Thermo Fischer Scientific, Waltham, MA, USA). In brief, 2 µL from each protein lysate was added to 2 µL of reagent A, followed by the addition of 100 µL of reagent B. After 30 min, the absorbance was read at 562 nm using an Infinite M200Pro plate reader (TECAN, Männedorf, Switzerland). In order to calculate a calibration curve, a dilution series of BSA (bovine serum albumin) was included in the measurement.

#### 4.6.5. Data Analysis (Metabolomics)

The data were exported and merged using an in-house written R script. The metabolites were considered valid when they appeared in a minimum of *n* = 3 biological replicates. The peak area of each metabolite was calculated by normalization to the internal standard cinnamic acid (and additionally to the protein content for in vitro studies). The technical variation of the GC-MS run is shown in Appendix A.

### 4.7. Proteom Profiling by Mass Spectrometry

#### 4.7.1. Protein Extraction and Digest

Cell pellets were resuspended in SDC buffer (1% sodium deoxycholate, 100 mM Tris-HCl pH 8, 1 mM EDTA, 150 mM sodium chloride 10 mM dithiothreitol, 40 mM chloroacetamide), heated for 10 min at 95 °C, cooled down to room temperature and incubated with 25 U Benzonase (Merck Burlington, MA, USA) for 30 min. The insoluble parts were pelleted by centrifugation for 20 min at 18,213× *g*, and the supernatant containing the protein extracts was collected. The protein concentration was measured (Bio-Rad DC Protein assay) and 100 µg of each sample was further processed using in solution digestion protocol. To each sample, 2 µg of sequence-grade trypsin (Promega, GmbH, Walldorf, Germany) and 2 µg of lysyl endopeptidase LysC (Wako Chemicals, Neuss, Germany) were added and incubated overnight at 37 °C. The reaction was stopped by adding trifluoroacetic acid (final concentration 1%) and the peptides were desalted and cleaned up using the StageTips protocol [42].

#### 4.7.2. LC-MS Analyses

Peptide samples were eluted from StageTips (80% acetonitrile, 0.1% formic acid), and after evaporating the organic solvent peptides were resuspended in a sample buffer (3% acetonitrile/0.1% formic acid). For each sample, two analytical runs were performed, injecting 1 µg of peptide material each time. The peptide separation was performed on a 20 cm reversed-phase column (75 µm inner diameter, packed with ReproSil-Pur C18-AQ (1.9 µm, Dr. Maisch GmbH, Ammerbuch, Germany) using a 200 min gradient with a 250 nL/min flow rate of increasing buffer B concentration (from 2% to 60%) on a High Performance Liquid Chromatography (HPLC) system (Thermo Fischer Scientific, Waltham, MA, USA). The peptides were measured on a Q Exactive HF-X instrument (Thermo Fischer Scientific, Waltham, MA, USA). The mass spectrometer was operated in the data dependent mode with a 60 K resolution, 3 × 10^6^ ion count target and maximum injection time of 10 ms for the full scan, followed by Top 20 MS2 scans with 15 K resolution, 1 × 10^5^ ion count target and maximum injection time of 22 ms. Each replicate was injected and measured twice.

#### 4.7.3. Data Analyses (Proteomics)

The raw data were processed using the MaxQuant software package (v1.6.3.4) [43]. The internal Andromeda search engine was used to search the MS^2^ spectra against a decoy human UniProt database (HUMAN.2019-07, with isoform annotations) containing forward and reverse sequences. The search included variable modifications of oxidation (M), N-terminal acetylation, deamidation (N and Q) and the fixed modification of carbamidomethyl cysteine. The minimal peptide length was set to six amino acids and a maximum of three missed cleavages was allowed. The FDR (false discovery rate) was set to 1% for peptide and protein identifications. Unique and razor peptides were considered for quantification. The retention times were recalibrated based on the built-in nonlinear time-rescaling algorithm. The MS^2^ identifications were transferred between runs with the “Match between runs” option, in which the maximal retention time window was set to 0.7 min. The integrated label-free quantification (LFQ) algorithm was applied.

The resulting text files were filtered to exclude reverse database hits, potential contaminants and proteins only identified by site. A statistical data analysis was performed using Perseus software (v 1.4.0.20). Biological replicates for each condition were defined as groups and the LFQ intensity values were filtered for a “minimum value of 3” per group. After log2 transformation, the missing values were imputed with random noise simulating the detection limit of the mass spectrometer. The imputed values are taken from a log normal distribution with 0.25× the standard deviation of the measured, logarithmized values, down-shifted by 1.8 standard deviations. The differences in protein abundance between the groups were calculated using a two-sample Student´s *t*-test. Proteins passing the FDR significance cut-off (0.05) were considered differentially expressed.

Perseus version 1.4.0.20 was used for clustering. The *Z*-score from the normalized peak areas from the technical and biological replicates for the annotated metabolites from the central carbon metabolism was calculated. Negative *Z*-scores were displayed in green and positive Z-Scores in purple. An unweighted average linking clustering and Euclidean distance preprocessed with k-means was used for hierarchical clustering.

### 4.8. AbsoluteIDQ p400 HR Assay and Sample Preparation

The Absolute*IDQ* p400 HR kit from Biocrates Life Science AG is a fully automated assay based on the phenylisothiocyanate (PITC) derivatization of the target analytes using internal standards for quantitation. Amino acids (21) and biogenic amines (21) are determined in liquid chromatography mass spectrometry (LC-MS) mode. Hexoses (1); neutral lipids like acylcarnitines (55); cholesteryl esters (14); diglycerides (18); and triclycerides (42); as well as polar lipids like phosphatidylcholines (172), lysophosphatidylcholines (24), sphingomyelins (31) and ceramides (9) are analyzed using flow injection analysis (FIA). The cell sample preparation was carried out according to the manufacturer’s protocol. Briefly, 20 µL of lysed cells and 10 µL of the internal standard was transferred to the upper 96-well plate and dried under a nitrogen stream. Thereafter, 50 µL of a 5% PITC solution was added to derivatize the amino acids and biogenic amines. After incubation, the filter spots were dried again before the metabolites were extracted using 5 mM of ammonium acetate in methanol (300 µL) into the lower 96-well plate for analysis after further dilution using the MS running solvent A. The quantification was carried out using internal standards (LC and FIA measurement) and a calibration curve (Cal 1 to Cal 7) only for LC measurement.

The evaluation of the instrument performance prior to the sample analysis was assessed by the system stability test (SST), as recommended by Biocrates. The separate test mixtures were provided with the kit for an LC-MS and FIA MS SST evaluation.

The LC-MS system was comprised of a 1290 Infinity II UHPLC-system (Agilent) coupled with a QExactive plus (Thermo Fisher Scientific, Waltham, MA, USA) in electrospray ionization (ESI) mode. The tune scan parameters were full MS with a scan range of 150.0 to 2000 m/z with a resolution of 70,000 in positive mode, with an automatic gain count (AGC) target 1e6 and a maximum injection time of 50. The sheet gas flow rate was 6, spray voltage 3.60 kV, capillary temperature 320 °C, S-lens RF level 60 and auxillary gas heater temperature 30 °C. The amino acids and biogenic amines were analyzed via LC-MS in positive mode with a runtime of 0 to 5.5 min; resolution of 70,000 or 3500 for LC1 and LC2, respectively; AGC target 1e6; 200 ms; and a scan range of 55 to 800 m/z. Twenty microliters of the sample extract were injected onto a Phenomenex 2.1 mm ID column (C18) protected by an Phenomenex SecurityGuardTM ULTRA Cartridge C18/XB-C18 at 50 °C using a 6.81 min solvent gradient employing 0.2% formic acid in water (solvent A) and 0.2% formic acid in acetonitrile (solvent B). Twenty microliters of the sample extract were used in the FIA in positive mode to capture Acylcarnitines, Glycerophospholipids and Sphingolipids, while the Hexoses were monitored in a subsequent run in positive mode. All the FIA injections were carried out using the Biocrates MS Running Solvent. For FIA 1, a runtime of 0 to 2.8 min in positive mode with resolution of 70,000, AGC target 3e6, maximum IT 250 ms and 8 scan ranges (1: 150 to 170 m/z; 2: 170 to 200 m/z; 3: 200 to 240 m/z; 4: 240 to 256 m/z; 5: 390 to 520 m/z; 6: 520 to 634 m/z; 7: 634 to 730 m/z; and 8: 730 to 931 m/z) were used. For FIA 2, a runtime of 0 to 2.8 min in positive mode with resolution of 70,000, AGC target 3e6, maximum IT 250 ms and 8 scan ranges (1: 256 to 280 m/z; 2: 280 to 305 m/z; 3: 305 to 335 m/z; 4: 353 to 363 m/z; 5: 363 to 390 m/z; 6: 390 to 415 m/z; 7: 415 to 445 m/z; and 8: 445 to 570 m/z.) were used. All the compounds were identified and quantified using isotopically-labeled internal standards and multiple reaction monitoring (MRM) for LC and full MS for FIA, as optimized and raw data were computed in Met*IDQ*^TM^ version Nitrogen (Biocrates Life Science AG, Innsbruck, Austria).

#### 4.8.1. Data Analysis (Biocrates Kit)

An in-house-developed script was used for the data quality analysis and preprocessing. The modification enabled the retention of metabolites considered above the limit of detection but below the limit of quantification by Biocrates software.

The metabolites were considered valid when they appeared in a minimum of three out of five biological replicates. Only analytes with values above the limit of detection (LOD) were considered. The LOD for individual analytes was defined as three times the median peak area in the blank samples (peak intensity was used for FIA data) or a minimum of 20,000 counts per second (cps). Analytes below the LOD were rejected. A 1.5 standard deviation of the total sample peak area per condition was calculated and used as the cutoff for sample outlier detection. The peak intensity was used for the FIA data. Finally, analytes with less than 3 values per condition were discarded from the downstream analysis.

## 5. Conclusions

In conclusion, we were able to reveal a progression-dependent metabolic pattern in the used OS cell line model and showed the importance of glutamine as an energy source. The data will help to provide new opportunities for personalized clinical therapy in OS patients.

## Figures and Tables

**Figure 1 cancers-12-01371-f001:**
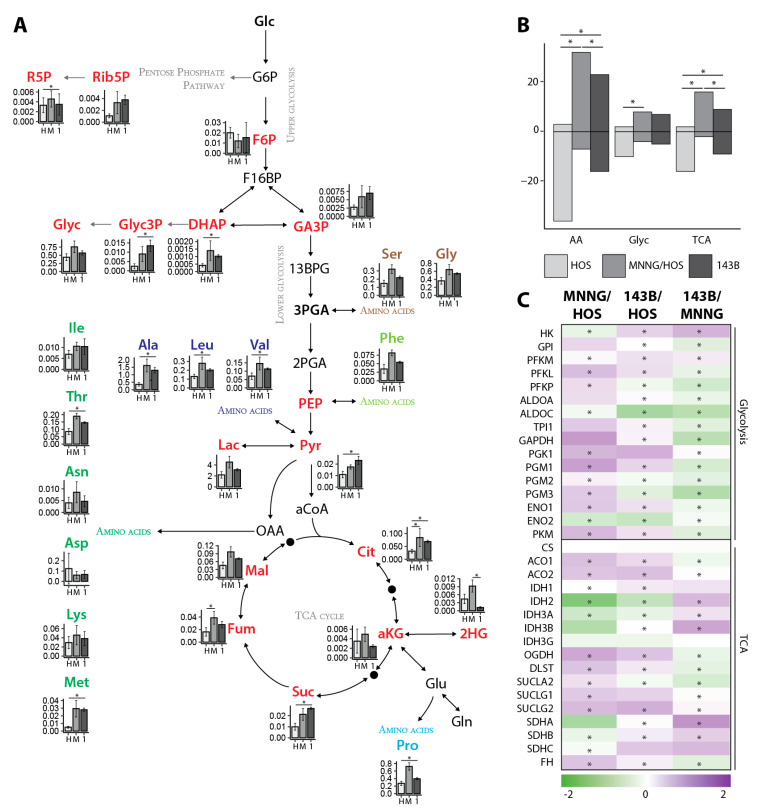
Enhanced metabolism in malignant and metastatic OS cells. (**A**) Levels of annotated metabolites of glycolysis, tricarboxylic acid (TCA) cycle, glycerol and amino acids in human osteosarcoma HOS (H), N-methyl-N’-nitro-N-nitrosoguanidine (MNNG)/HOS (M) and 143B (1) cells (each comprising *n* = 3 biological and *n* = 2 technical replicates). Data shows the mean and standard deviation of the normalized peak areas. Samples were analyzed using an unpaired Student’s *t*-test, with a *p* ≤ 0.05 deemed statistically significant and indicated with a star (*). (**B**) Number of metabolites after unit scaling of normalized peak areas (in total and according to the subsets of metabolic pathways comprising glycolysis (glyc), TCA and amino acids (AA). Metabolites with higher peak areas than the average for that metabolite are shown above zero and those with lower peak areas below. We used the Pearson’s chi-squared test to perform a chi-squared contingency table test and goodness-of-fit with a threshold of *p* < 0.05 for statistically relevant differences between the metabolic groups and the total number of metabolites (indicated with a star (*)). Y-axis: frequency of metabolites above or below individual metabolite mean average. (**C**) Heat map from proteins associated with the central carbon metabolism comparing MNNG/HOS to HOS, 143B to HOS and 143B to MNNG/HOS (MNNG) cells (each *n* = 5 replicates). Ratios from the log2 values of the label-free intensities from the biological replicates are shown. Samples were analyzed using an unpaired Student’s *t*-test with a *p* < 0.05 deemed statistically significant and indicated with a star (*). 2HG: 2-hydroxy-glutaric acid. 2PGA: glyceric-acid-2-phosphate. 3PGA: glyceric-acid-3-phosphate. 13BPG: 1/3-bis-phosphoglyceric acid. aCoA: acetyl-CoenzymeA. ACO: aconitase. Ala: alanine. ALDO: aldolase. aKG: alpha-ketoglutaric acid. Asn: asparagine. Asp: aspartic acid. Cit: citric acid. CS: citrate synthase. DHAP: dihydroxyacetonephosphate. DLST: dihydrolipoamide succinyltransferase (E2) component of the 2-oxoglutarate dehydrogenase complex. ENO: enolase. F6P: fructose-6-phosphate. F16BP: fructose-1/6-bisphopsphate. FH: fumarate hydratase. Fum: fumaric acid. G6P: glucose-6-phosphate. GA3P: glyceraldehyde-3-phosphate. GAPDH: glycerinaldehyde-3-phosphate dehydrogenase. Glc: glucose. Gln: glutamine. Glu: glutamic acid. Glyc3P: glycerol-3-phosphate. Gly: glycine. Glyc: glycerol. GPI: glucose-6-phosphate isomerase. HK: hexokinase. IDH: isocitrate dehydrogenase. Ile: isoleucine. Lac: lactic acid. Leu: leucine. Lys: lysine. Mal: malic acid. Met: methionine. OAA: oxaloacetate. OGDH: 2-oxoglutarate dehydrogenase (E1) component of the 2-oxoglutarate dehydrogenase complex.Phe: phenylalanine. PEP: phosphoenol-pyruvic acid. PFK: phosphofructokinase. PGK: phosphoglyceratkinase. PGM: phosphoglycerate mutase. PKM: pyruvate kinase. Pyr: pyruvic acid. Pro: proline. SDH: succinate dehydrogenase. Ser: serine. Suc: succinic acid. SUCL: succinyl-CoA synthetase. TCA: tricarboxylic acid. Thr: threonine. TPI: triosephosphatisomerase. Tyr: tyrosine. Val: valine. R5P: ribose-5-phosphate. Rib5P: ribulose-5-phosphate.

**Figure 2 cancers-12-01371-f002:**
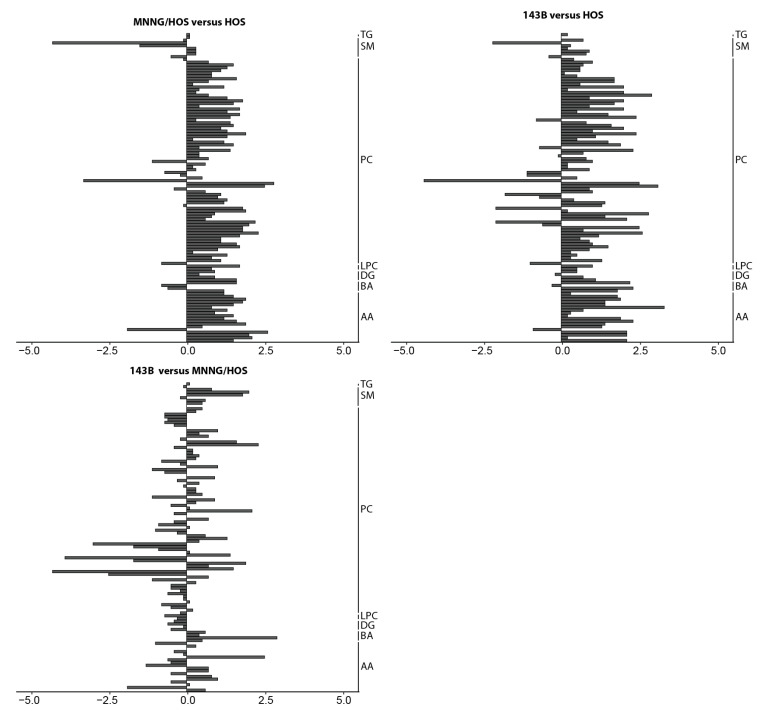
Enhanced lipid metabolism in malignant and metastatic osteosarcoma (OS) cells. Measured compounds comparing MNNG/HOS to HOS, 143B to HOS and 143B to the MNNG/HOS cells. For each cell type, *n* = 5 replicates were measured. Data are log2 values from the ratios calculated from the mean of the normalized areas (liquid chromatography measurement) or intensities (flow injection analysis measurement). AA: amino acids. BA: biogenic amines. DG: diglycerides. LPC: lysophosphatidylcholines. PC: phosphatidylcholines. SM: sphingomyelins. TG: triglycerides.

**Figure 3 cancers-12-01371-f003:**
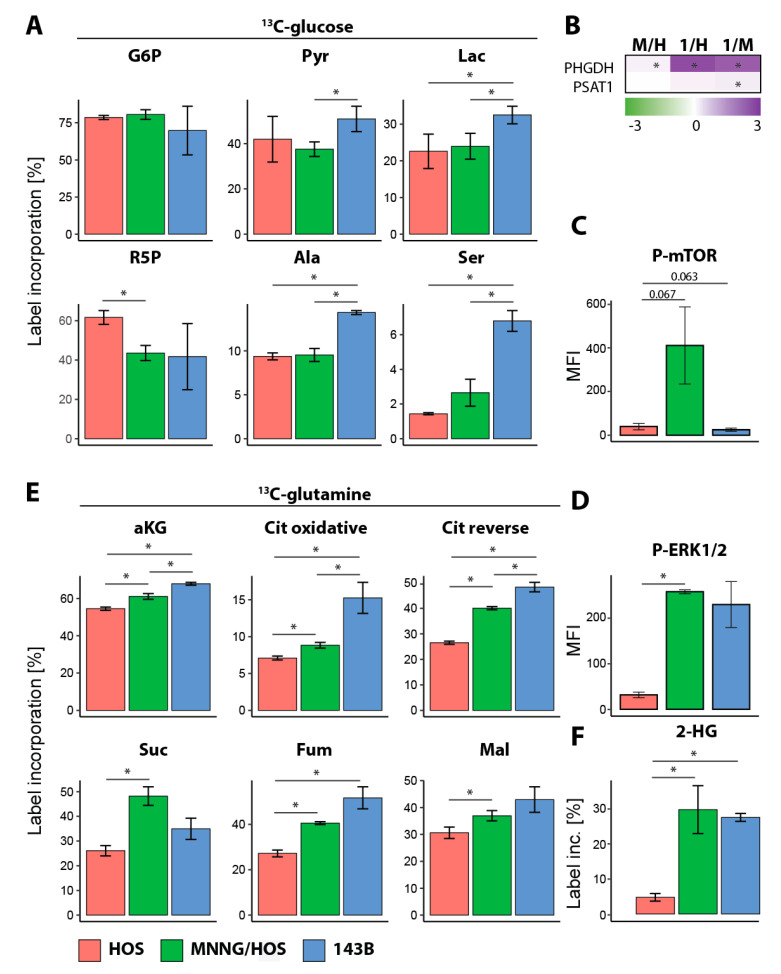
Increased metabolic flux cycle in metastatic OS cells. (**A**,**E**,**F**) Label incorporation of glycolytic intermediates in OS cells in the presence of ^13^C-glucose for 30 min (**A**) or ^13^C-glutamine for 60 min (**E**,**F**). Each *n* = 3 replicates. (**B**) Log2 label-free intensities were used to calculate the ratio of MNNG/HOS (M) to HOS (H), 143B (1) to HOS and 143B to the MNNG/HOS cells. Each *n* = 5 replicates. (**C**,**D**) Mean fluorescence intensity (MFI) of phosphorylated mechanistic target of rapamycin (P-mTOR) and phosphorylated extracellular signal related kinases 1 and 2 (P-ERK1/2). Each *n* = 3 replicates. Samples were analyzed using an unpaired Student’s *t*-test with a *p* ≤ 0.05 deemed statistically significant (indicated by a star (*)). 2-HG: 2-hydroxy-glutaric acid. Ala: alanine. aKG: alpha-ketoglutaric acid. Cit: citric acid. Fum: fumaric acid. G6P: glucose-6-phosphate. Lac: lactic acid. Mal: malic acid. PHGDH: D-3-phosphoglycerate dehydrogenase. PSAT1: phosphoserine aminotransferase. Pyr: pyruvic acid. R5P: ribulose-5-phosphate. Ser: serine. Suc: succinic acid.

**Figure 4 cancers-12-01371-f004:**
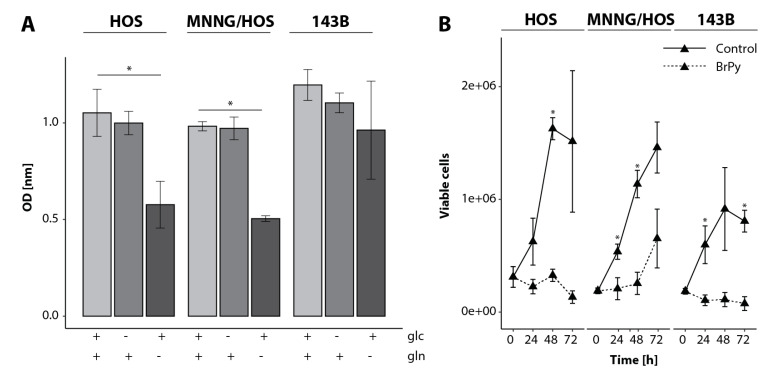
Progression-dependent switch in nutrient sources. (**A**) Viability of osteosarcoma cell lines HOS, MNNG/HOS and 143B after glucose (glc) or glutamine (gln) starvation for 48 h. For each condition, *n* = 4 replicates were measured. (**B**) Cells were treated with 0.25 mM 3-Bromopyruvic acid (BrPy) or control solvent phosphate buffered saline. For each condition, *n* = 3 replicates were measured. Samples were analyzed using an unpaired Student’s *t*-test, with a *p* ≤ 0.05 deemed statistically significant (indicated by the star (*)). OD: optical density.

**Figure 5 cancers-12-01371-f005:**
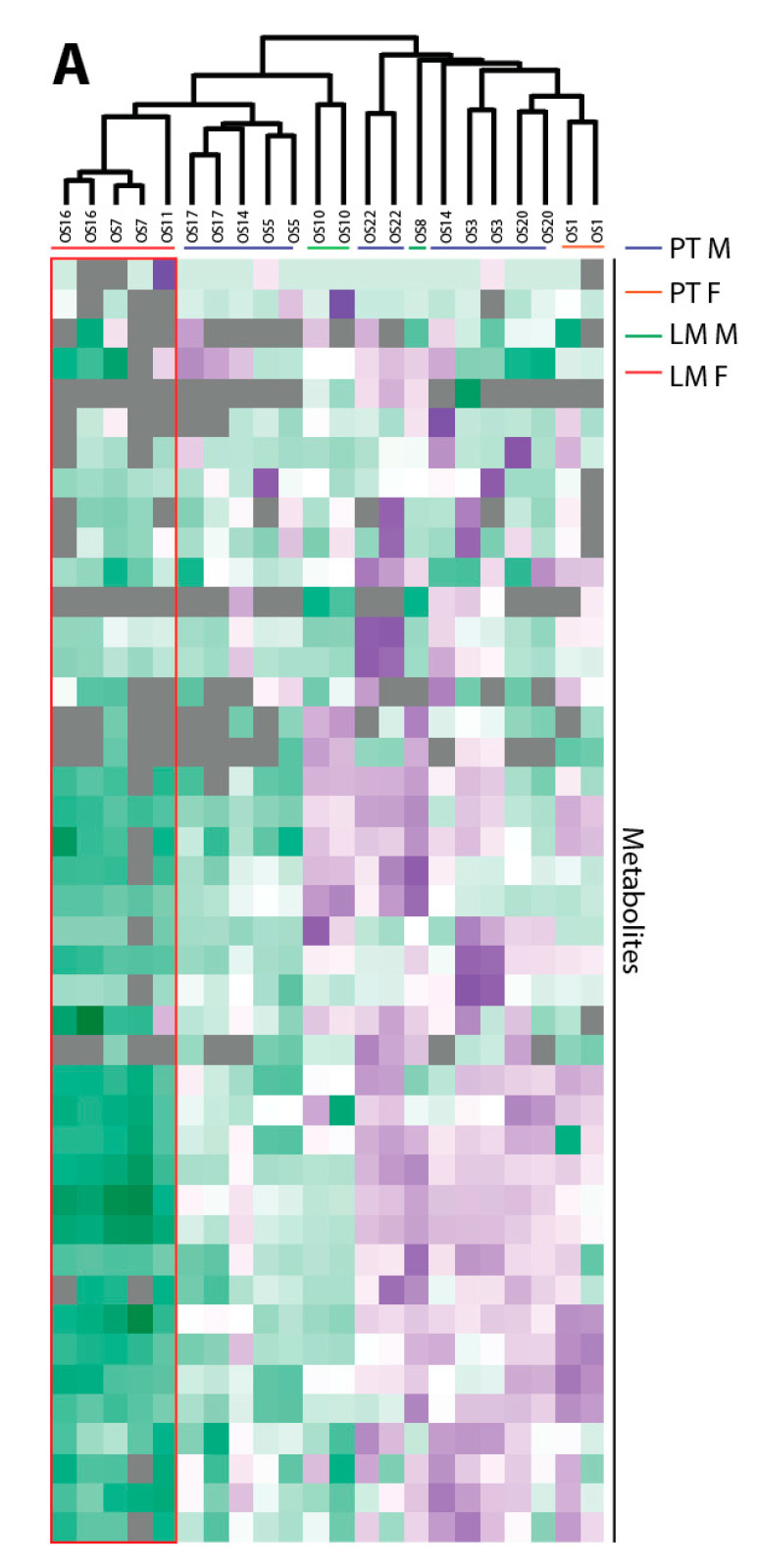
Sex-dependent metabolic alteration in OS patients with primary tumors (PT) and lung metastases (LM). (**A**) Heat map for the unsupervised hierarchical clustering of the *n* = 7 PT and *n* = 5 LM (males = M and females = F). Perseus version 1.4.0.20 was used for clustering. The Z-score from the normalized peak areas from the technical and biological replicates for the annotated metabolites from the central carbon metabolism was calculated. Negative Z-scores were displayed in green, positive *Z*-Scores in purple. An unweighted average linkage clustering and Euclidean distance preprocessed with k-means was used for the hierarchical clustering. (**B**) Levels of annotated metabolites of glycolysis, the TCA cycle, glycerol and amino acids in *n* = 7 PT and *n* = 5 LM from males and females. Data show the mean and standard deviation of the normalized peak area. Samples were analyzed using an unpaired Student’s *t*-test, with a *p* ≤ 0.05 deemed as statistically significant and indicated by the star (*). 2HG: 2-hydroxy-glutaric acid. 2PGA: glyceric-acid-2-phosphate. 3PGA: glyceric-acid-3-phosphate. 13BPG: 1/3-bis-phosphoglyceric acid. aCoA: acetyl-CoA. Ala: alanine. aKG: alpha-ketoglutaric acid. Asn: asparagine. Asp: aspartic acid. Cit: citric acid. Cys: cysteine. DHAP: dihydroxyacetonephosphate. F6P: fructose-6-phosphate. F16BP: fructose-1/6-bisphopsphate. Fum: fumaric acid. G6P: glucose-6-phosphate. GA3P: glyceraldehyde-3-phosphate. Glc: glucose. Gln: glutamine. Glu: glutamic acid. Glyc3P: glycerol-3-phosphate. Gly: glycine. Glyc: glycerol. Ile: isoleucine. Lac: lactic acid. Leu: leucine. Lys: lysine. Mal: malic acid. Met: methionine. OAA: oxaloacetate. Phe: phenylalanine. PEP: phosphoenol-pyruvic acid. Pyr: pyruvic acid. Pro: proline. Ser: serine. Suc: succinic acid. TCA: tricarboxylic acid. Thr: threonine. Trp: tryptophan. Tyr: tyrosine. Val: valine.

**Table 1 cancers-12-01371-t001:** Osteosarcoma patient’s characteristics of primary tumors (PT) and lung metastases (LM). *: systemic chemotherapy. F: female. ID: identification. M: male. NA: not available.

ID	Sex	Age	Grade	Chemotherapy	Responder	Metastasis	Type
OS1	F	15	3	Yes	No	NA	PT
OS14	M	56	3	Yes	Yes	Yes	PT
OS15	F	74	2	No	-	NA	PT
OS17	M	58	3	Yes	No	NA	PT
OS20	M	38	3	No		No	PT
OS22	M	23	1	No		No	PT
OS7	F	22	3	Yes *		-	LM
OS8	M	67	3	No		-	LM
OS10	M	24	3	Yes	No	-	LM
OS11	F	46	3	Yes		-	LM
OS16	F	38	2	Yes	Yes	-	LM

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
