# Peer review of "Progression-Dependent Altered Metabolism in Osteosarcoma Resulting in Different Nutrient Source Dependencies"

_cancers, 2020, doi:10.3390/cancers12061371_

Round 1

Reviewer 1 Report

The authors present interesting work concerning progression-dependent metabolic differences found in osteosarcoma model in vitro and in vivo. The manuscript is generally well written. 

However, the discussion is not sufficiently described. The impact of amino acids tumor micro-enviroment, e.g. glutamine, glycine, serine on aggresiveness of osteosarcoma cells, should be more widely discussed.

Moreover, the amount of patients  is limitation of this study. Therefore the conclusions concerning sex differences needs to be further confirmed on higher amount of human subjects. This statement should be clearly indicated. The inclusion and exclusion criteria to this study should be clearly described. 

Author Response

Response to Reviewer 1 Comments
We thank the reviewer for their constructive comments and feedback. We agree that the amount of patient samples is a limitation criteria for the study. We now included a clear statement that this needs to be confirmed on higher amount of samples.

Changes were made according to reviewer’s suggestions. We hope that the improvements we have made on your recommendation have improved the paper. We improved the description of the methods part.
We noted the reviewers concerns about English language. We could not find any major language issues, including our native English speakers, but we have extensively edited the paper to improve clarity in places.

Major concerns:

Point 1: However, the discussion is not sufficiently described. The impact of amino acids tumor micro-enviroment, e.g. glutamine, glycine, serine on aggressiveness of osteosarcoma cells, should be more widely discussed.

We thank the reviewer for the points made. Since the microenvironment of tumors is an important factor during metastasis, we extended the discussion regarding the impact of amino acids in the tumor environment and aggressiveness (bold text).

Line 308: “It is already known that targeting amino acid metabolism in cancer is a promising strategy for the development of novel therapeutic agents. Amino acids are essential to support the high metabolic demands of tumor cells to deal with the conditions of the tumor microenvironment. Especially metastatic cells need to deal with certain stressful conditions in the tumor microenvironment. The amino acid arginine is an important precursor for the synthesis of proteins, urea, creatinine and for the synthesis of glutamate, nitric oxide and agmatine [29]. A lot of tumors depend on exogeneous arginine for their growth since they lack argininosucchinate synthetase 1 (ASS1) [30]. OS patients lacking ASS1 correlates with the development of pulmonary metastases [31]. Serine and glycine metabolism are interconnected via the glycine cleavage system, a major metabolic pathway in one-carbon metabolism that provides cofactors for pyrimidine and purine nucleotide biosynthesis. Cancer cells use both amino acids to synthesize building material for cell growth and proliferation [32]. The serine/glycine metabolism controlled by the mTORC1 pathway act as a protective system in OS cells [33]. This metabolic signaling promotes OS cell proliferation and the ability to deal with micro-environmental stress leading to enhanced OS cell survival. It is known that cancer cells have a higher capacity for de novo serine synthesis via the PHGDH pathway. The serine biosynthetic pathway is upregulated in highly metastatic breast cancer cells and associated with poor survival [34]. In our study glycine and serine showed significantly lower pools in female OS LM compared to male LM and PT and in the counterpart in vitro cellular models 143B compared to MNNG/HOS. However, the in vitro flux analysis revealed a high label incorporation into serine. In accordance, a significant increase in the serine pathway proteins were found, suggesting that serine and glycine pools were decreased due to increased use as an energy source.
Metastatic OS cells relay on glutamine as a key source for proliferation [35].”

Point 2: Moreover, the amount of patients is limitation of this study. Therefore the conclusions concerning sex differences needs to be further confirmed on higher amount of human subjects. This statement should be clearly indicated. The inclusion and exclusion criteria to this study should be clearly described.

We thank the reviewer for their observation and we entirely agree. This is a rare disease and samples are difficult to obtain. We have included a clear statement that a larger sample sizes for both in vivo and in vitro progression studies are required to confirm the results:

Line 380: “However, further follow up with a larger patient sample size and multiple male and female progression-dependent OS cell line models is required.”

The material was selected upon availability. OS is a rare disease and we only have limited access to the tumor samples. No exclusion or inclusion criteria above and beyond diagnosed OS was used due to the rarity and difficulty of collecting samples.

Line 389: “Inclusion criteria was a histo-pathological diagnosed OS.”

Reviewer 2 Report

This study is technically sound except that studying metabolism in cells cultured in 21% O2 could be completely irrelevant. The aim was to better document the “metabolic transformation during metastasis”. In contrast with the statement in the abstract many data on osteosarcoma metabolism have been produced and already served to classify tumors. Modifications of cell proliferation/death were in line with previous reports. The novelty may be the comparison of cells with malignant vs metastatic capacities. However, something appears very confusing do the authors want to address metabolism associated to metastatic capacities of cells in the primary tumors or modifications of metabolism during metastatic process or metabolism of metastasis cells in the metastatic site. This has to be clarified.

Also the way the tumors were clustered has to be better explained.

To support the idea of a potential sex-dependent metabolic alteration the authors have to compare a cell line derived from a male patient and to explore data bases countaing metabolomic data to compare a larger number of tumors.

Finally, one point should be discussed is how to include the heterogeneity of tumors and cell lines as a parameter that impacts data interpretation.

In conclusion, major modifications appear required for publication including writing a clearer abstract, add some explanations about tumor analyses and clustering. Repeat at least some analyses concerning the metabolic flux with a cell line derived from a male.

Author Response

Response to Reviewer 2 Comments
We thank the reviewer for their constructive comments and feedback. Changes were made according to reviewer’s suggestions. We hope that the changes we have made on your recommendation have improved the paper.

Major concerns:

Point 1: In contrast with the statement in the abstract many data on osteosarcoma metabolism have been produced and already served to classify tumors.

Compared with more common cancers, the current metabolomics research on OS is relatively modest. However, we have reflected the current research better by including a statement in our text to this effect and updating our references.

Line 21: “Alterations to the metabolome and its transformation during metastasis aids understanding of mechanism and provides information on treatment and prognosis.”
Line 58: “Previous research in OS is more modest and has focused on understanding the pathogenesis and development of OS. Metabolic alterations have previously been used to classify OS tumors [8-16]. However, there are still unanswered questions relating to the metabolic transformation during metastasis.”

Point 2: However, something appears very confusing do the authors want to address metabolism associated to metastatic capacities of cells in the primary tumors or modifications of metabolism during metastatic process or metabolism of metastasis cells in the metastatic site. This has to be clarified.

The authors did a comparison of different progression steps in vitro to see metabolic patterns specifically due to the different malignancy of the OS cells. The analysis of the patient tumors was done to see metabolic patterns which distinguished primary and metastatic OS tumors. Due to the rarity of the disease we were not able to source tumor tissue from the same patients. The following chapter is removed due to confusion about then comparison done in vitro and in vivo.

Line 266: “To combine the results from our in vivo and in vitro approach a comparison of malignant and PT as well as metastatic and LM could be made. Interestingly, the parental HOS cell line is cultured from a female patient. Therefore, the in vitro results of a decreased CCM in metastatic 143B cells are in line with the observations from the female lung metastasis patients.”
Line 281: “…related to OS metastasis…” was changed to “…from primary tumors and lung metastases…”

Point 3: Also the way the tumors were clustered has to be better explained.

We thank the reviewer for the point made. The cluster criteria were described in more detail in the Figure Reference and material and methods part.

Line 241/541: “Perseus version 1.4.0.20 was used for clustering. The Z-score from the normalized peak areas from the technical and biological replicates for the annotated metabolites from the central carbon metabolism was calculated. Negative Z-scores were displayed in green, positive Z-Scores in purple. An unweighted average linkage clustering and Euclidean distance preprocessed with k-means was used for hierarchical clustering.”

Point 4: To support the idea of a potential sex-dependent metabolic alteration the authors have to compare a cell line derived from a male patient and to explore data bases countaing metabolomic data to compare a larger number of tumors.

We agree, but unfortunately such a model is not currently commercially available. There is only a noncommercial highly metastatic cell line male available (published by Yu et al. 2019; 10.21037/atm.2019.09.23) but this has no benign and malignant counterpart. Similarly, a non-commercial male metastatic cell line called MG63.3 exists (Khanna et al. 10.1023/a:1006767007547), but only with the benign counterpart MG63. Since we saw the main differences of decreased metabolic pools and increased metabolic flux comparing malignant and metastatic cell lines, this model can`t be used. To exclude genetic backbone differences a similar model like the HOS cells are necessary for a direct comparison. In addition, multiple models for both male and female would be required to draw sound conclusions. Since osteosarcoma is a rare disease we had only limited availability to the patient material. We are currently forming a cooperation with others interested in OS to combine resources and collect the necessary cell models and patient samples to take this research further.

Point 5: Finally, one point should be discussed is how to include the heterogeneity of tumors and cell lines as a parameter that impacts data interpretation.

The reviewer highlighted a good point. We now included the discussion about tumor and cell heterogeneity.
We have previously carried out very limited studies on tumor heterogeneity and shown that for one sample it was less than the biological variability between tumor samples. We hope to carry out more extensive studies on this if we are able to get access to a larger cohort. Meanwhile, we have recognized that this issue should be directly addressed and so have added a line in the discussion:

Line 359:” OS is one of the most complex oncological diseases in terms of genetic aberrations [38,39]. Tumor heterogeneity relays on the cancer cells and the tumor microenvironment which is composed of different cells types including immune cells, mesenchymal stem cells, endothelial cells and fibroblast. To reduce the genetic variation for data analysis, a progression model was used that was derived from a single cell line. To statistically verify the obtained results and to evaluate sex-specific differences in progression states would ideally require more of these progressive OS cell lines from different sexes. Further follow up would also allow us to better understand the effects that tumor heterogeneity may exert on the results.”

Round 2

Reviewer 1 Report

The authors corrected the manuscript as suggested.

Reviewer 2 Report

In spite of the quantity of data provided, this work fails to bring significant insights about changes in metabolism that may drive or go with the metastatic process. The conclusion about the possible difference between male and female is more speculative than really supported by the results. In conclusion, these results appear very preliminary.